# Integrated 0.35-µm CMOS Control Circuits for High-Performance Voltage Mode DC–DC Boost Converter

Chan-Soo Lee ⬤, Munkhsuld Gendensuren, Bayarsaikan Dansran, Bierng-Chearl Ahn *⬤ and Seong-Gon Choi *

School of Electrical and Computer Engineering, Chungbuk National University, Cheongju 28644, Republic of Korea
* Correspondence: bician@chungbuk.ac.kr (B.-C.A.); sgchoi@chungbuk.ac.kr (S.-G.C.);
Tel.: +82-43-261-3194 (B.-C.A.)

**Abstract:** The integrated DC–DC converter is appropriate for use in many domains, namely, display, cellular, and portable applications. This paper presents an integrated control circuit for a monolithic voltage mode DC–DC boost converter for display driver applications. The control circuits consist of a transconductance amplifier, a comparator, and an oscillator. The boost converter consists of an inductor, two MOSFET, and an output *RC* filter. The control circuits are designed for fast transient response and low output ripple. The transconductance amplifier, comparator, and oscillator in the control circuit are designed to operate at a supply voltage of 3.3 V and an operating frequency of 5.5 MHz. The transconductance amplifier consists of an operational amplifier and an *RC* filter in the feedback path. The *RC* filter has a pole with a sufficient phase margin for high stability. The control circuits are realized in a 0.35-µm CMOS process together with the DC–DC converter. The fabricated DC–DC converter was evaluated by experiment and simulation. Testing of the proposed control circuits shows that the output transient time can be controlled within 7 µs, and the output voltage is accurately controlled with a ripple ratio of 3%.

**Keywords:** CMOS; DC–DC converter; control circuit; integration; voltage–mode; boost converter





## 1. Introduction

In this paper, we present a low-power system on a DC–DC boost converter for display driver applications. There have been intensive research efforts on CMOS-integrated DC–DC boost converters for low-power applications. Lai and co-workers have proposed a 0.18-um CMOS device design of a 0.6–1.7 V boost converter using a 2.3-GHz oscillator and two differential cross-coupled rectifiers [1]. Chang and co-workers have a single-inductor multi-output (SIMO) DC–DC converter 0.18-um 1P6M CMOS process for converting 1.8 V to 1.5, 2.0, 2.2, 2.5 V [2]. They employed two comparator loops and two error amplifier loops. Jiang and co-workers investigated a buck-boost converter with 0.22–2.4 V input and 0.85–1.2 V output employing an algorithmic voltage-feed-in topology [3]. Choi and co-workers presented a soft-switching hybrid DC–DC converter realized in a 65-nm CMOS process [4]. They achieved 79.5% efficiency. Guierrez proposed a fully-integrated DC–DC converter for IoT power supply applications [5]. He achieved 69% peak efficiency with 180-nm bipolar CMOS-DMOS technology. Song and co-workers studied a switched-capacitor DC–DC step-up converter for implantable neural interface applications [6]. They achieved 82.6% efficiency with a 180 nm CMOS 1P6M process.

In DC–DC converters, the converter consists of a power stage and a control stage. The control stage requires an op-amplifier [7–9] for low-power operation and high accuracy. The op-amplifier accurately controls the drain-source voltage of the power transistor. This control circuit [10–12] is a high-gain amplifier in a differential pair.

In this paper, we propose integrated control circuits for a high-performance DC–DC boost converter for display applications. While current mode control [13,14] is much



better than voltage mode control, there are a lot of difficulties in the implementation of the current sensing circuit. In a voltage-mode DC–DC converter, the performance of the voltage control circuit is crucial to the converter's performance. The aim of this paper is to develop fully integrated control circuits for high-performance DC–DC boost converters for 3.3-V input and 5 to 7-V output voltage operation. The proposed control circuits work on the pulse width modulation principles and consist of an operational transconductance amplifier (OTA) or compensator, a comparator, a 5.5-MHz oscillator, and a gate driver. In the design of the control circuits, the main focus is on a fast response to load change or a short transient time and a low ripple in the output voltage. Each sub-circuit in the control circuits is designed for the overall objectives of fast response and low ripple voltage based on the 0.35-μm CMOS process.

The structure of the proposed voltage-mode boost converter consists of a power stage and a control stage, as shown in Figure 1. A converter usually consists of a power switching stage and a feedback control circuit. Among the variable circuit elements of voltage-mode converters, a transconductance amplifier, a comparator, and an oscillator in the control circuits are critical to the performance of the overall feedback operation, which requires fast dynamic response and reduction in sub-harmonic oscillation. The feedback network is a system-on-integrated circuit for voltage-mode gate driver control. By using the properties of the CMOS transistor, a high-performance control circuit can be manufactured. It enables accurate sensing of the inductor current at high frequency.

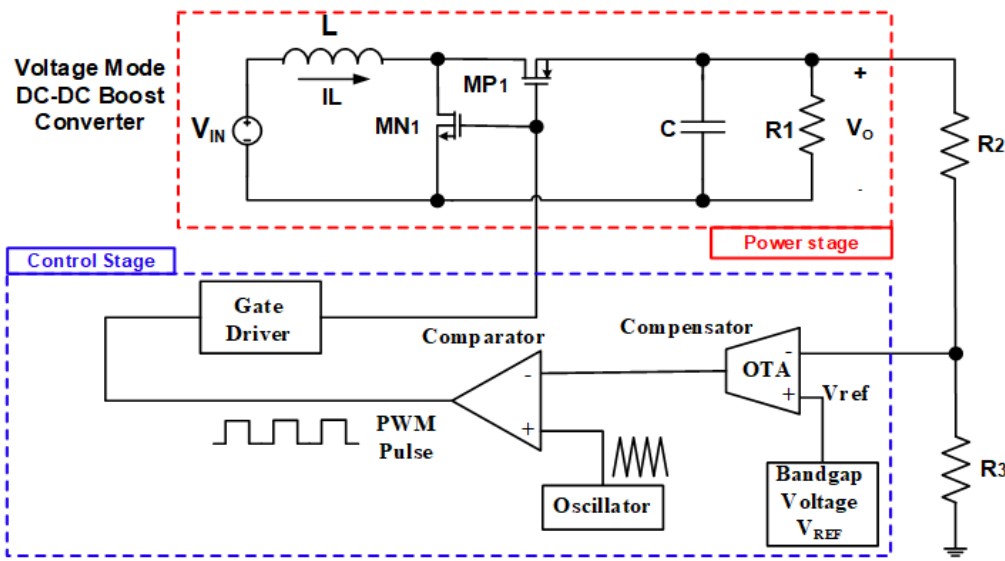

**Figure 1.** Block diagram of the proposed voltage–mode boost DC–DC converter.

The power stage includes two MOS switches $MN_1$, $MP_1$ with an off-chip *LC* filter. The voltage-mode DC–DC boost converter in a system-on-chip configuration is designed with a 0.35-μm CMOS process for low-power operation and full integration of the circuitry. The input voltage of the op-amplifier is scaled down by $R_2$ and $R_3$. The off-chip *LC* filter is designed for inductance of 1–10 μH and capacitance of 0.1–1 μF. The control circuit is designed for a supply voltage of 3.3 V and an operating frequency of 5.5 MHz. The operation of the proposed control circuits with the voltage-mode DC–DC boost converter was verified by simulation and by experiments.

## 2. Control Circuits

### 2.1. Operational Transconductance Amplifier (OTA) Circuit

The two-stage operational amplifier configuration is a popular structure for CMOS op-amps, and it has reasonably good quality in addition to the simplicity of the circuit. As seen in Figure 2, the two-stage operational amplifier configuration is composed of a

differential amplifier input stage, current sources, and a feedback compensator, which requires a fast-transient response with low-power consumption.

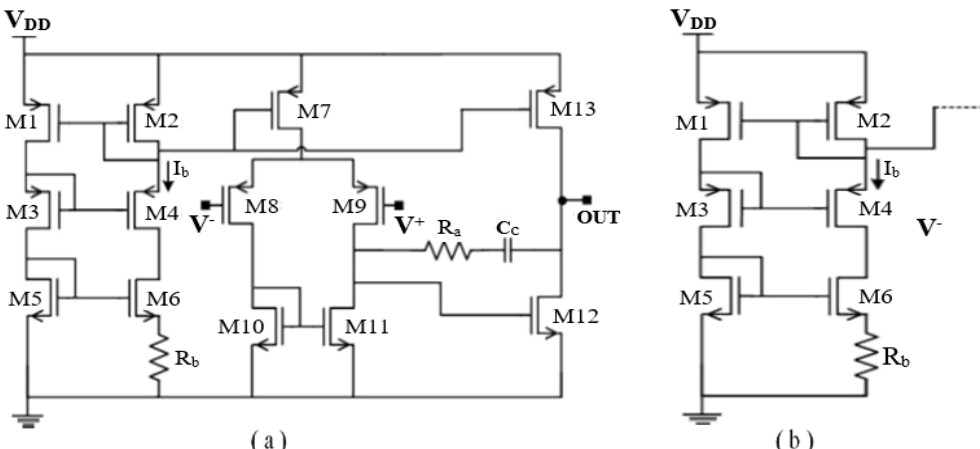

**Figure 2.** Proposed OTA circuit: (**a**) two–stage CMOS op–amp configuration; (**b**) configuration used to bias two–stage op–amp.

The OTA in Figure 2a consists of an input differential stage, a source follower, and current mirrors. A compensator is included in the op-amp to increase the stability of the frequency response and to achieve a fast response time. A differential amplifier is used in the first stage since it is less sensitive to noise due to its high common mode rejection ratio (CMRR). Although there is only one capacitor $Cc$, in the negative feedback path of the second stage (frequency-compensated) in the OTA, a resistor $R_a$ in series with $C_C$ is used to improve the phase margin of the op-amp by placing zero in the negative axis. It usually includes a compensation circuit to achieve stability of frequency response and fast response time. The compensator is used to realize poles or zeros for a sufficient phase margin for high stability. A drawback of this configuration is that it does not have a low output resistance suitable for driving low input impedance loads. In order to implement $I_{REF}$ current in Figure 2a, the bias circuit in Figure 2b needs to be capable of providing current independent of the supply voltage and the MOSFET threshold voltage. A useful and interesting property of the bias circuit is that the transconductances of the transistors biased by this circuit are only dependent on $R_b$ value and device dimensions. As seen in Figure 2b, a resistor $R_b$ is connected in series with the source of $M_6$. This resistor $R_b$ is important in determining the bias current $I_b$ and the transconductance of the transistor $M_6$.

Figure 3a is an operational amplifier with a compensator. The transfer function of the compensator with poles due to the $RC$ filter [15,16] can be written in the following form by applying Miller's theorem. The resistors and capacitors are used to create poles or zeros in the compensator, where $A_{op}$ and $R_{out}$ are the gain and output resistance of the op-amp, and $R_1$ and $C_1$ are the resistance and capacitance of the compensator.

$$\frac{V_{control}}{V_{out}} = \frac{k}{(1 + sR_1C_1)} \tag{1}$$

$$f_C = \frac{1}{2\pi R_1 C_1} \tag{2}$$

$$A_S = \frac{A_{OP}}{(1 + S/\omega_P)}) \tag{3}$$

$$\omega_P = [R_P{\cdot}C_P]^{-1} \tag{4}$$

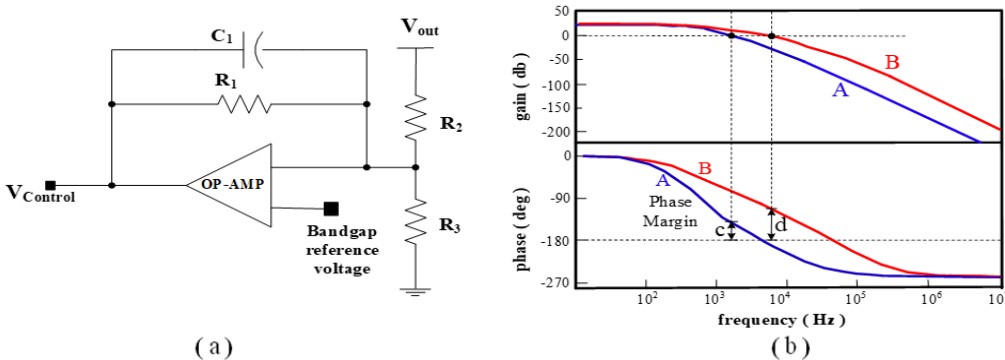

**Figure 3.** Proposed OTA circuit: (**a**) schematic of compensator with operational amplifier; (**b**) frequency response of the loop gain (A: without pole–zero optimization, B: with pole–zero optimzation).

The stability and frequency response are determined directly by the pole. Whether the converter circuit is stable or not can be determined by examining the loop gain as a function of the frequency, where $k$, $R_1$, and $C_1$ are the gain, resistance, and capacitance in the compensator. The $f_C$ is the compensating frequency determined by the resistance $R_1$ and capacitance $C_1$. Figure 3b shows the gain of the compensator versus the frequency, where phase $B$ with pole-zero optimization is shown together with that of case $A$ without pole-zero optimization. In case $A$, the phase margin ($c$) is almost 0° while the phase margin ($d$) is 47°. The frequency characteristic of the compensated op-amplifier is combined into the control feedback loop characteristics of the DC–DC boost converter. The results indicate that the compensator frequency should be higher than the initial unoptimized frequency for increased phase margin and, thus, higher stability. Another factor affecting phase margin and stability is the gain of the op-amplifier given in Equation (3). The op-amplifier is designed to maintain a small quiescent current during normal operation and to withstand large input signal fluctuations. The transconductance of the op-amplifier was well controlled to maintain the stability of the converter.

*2.2. Comparator Circuit*

The comparator [17,18] in Figure 4a is for the pulse-width modulation (PWM) control. It is composed of a bias circuit, an input differential stage, and a latch. The bias circuit is almost the same as that of the op-amplifier. The input stage is a circuit for start-up and bandgap reference. A start-up circuit is added to ensure that the bandgap reference circuit turns on when the supply voltage is applied. The load circuit employs a current mirror configuration and thus presents an amplifier with a high-resistance load. With this approach, a gain larger than 20 is achievable. The inverter and latch are used for a clear logic response and can act as a driver stage, so transistors in the current mirror can be made smaller for reduced parasitic capacitance.

The operation of the comparator is a simple process. When the compensator op-amplifier output voltage is less than that of the 5.5 MHz triangular waveform voltage, the output voltage of the comparator is low. When the compensator output voltage is greater than the triangle waveform voltage, the output of the comparator is high. Figure 4b shows the output waveforms of the op-amplifier (A) and the comparator (B). This is obtained from the Cadence simulation tool of the block diagram containing the power and control stage circuits. The output of the op-amplifier (A) quickly settles down due to the high conductance $g_m$ of the differential pair. The comparator repeats the digital high and low logic until the op-amp signal (A) passes the transient response.

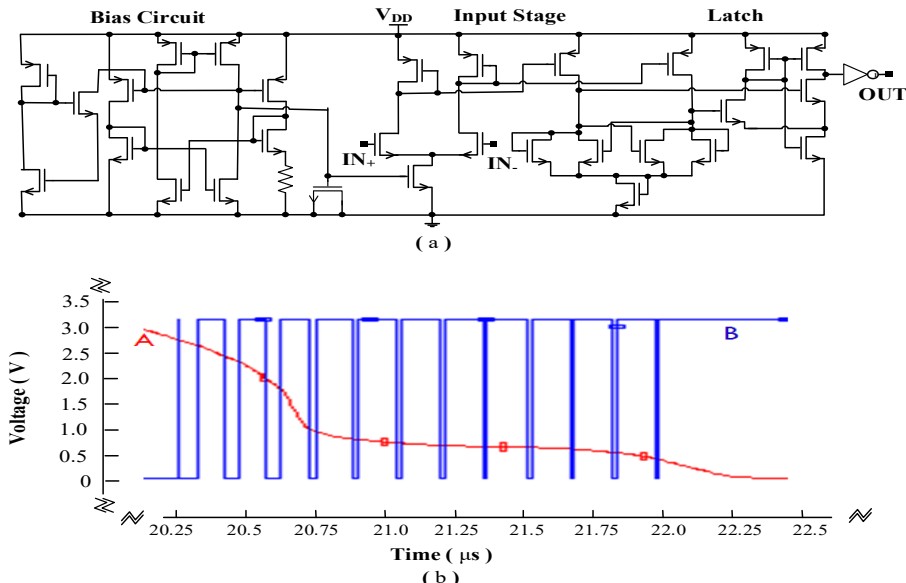

**Figure 4.** Proposed comparator circuit: (**a**) comparator circuit; (**b**) outputs of the op-amplifier (A) and comparator (B).

## 2.3. Oscillator Circuit

An oscillator signal is used for PWM control and for PWM switching [12]. The switch control signal is generated by comparing the compensator op-amp output voltage with a triangular waveform obtained from the oscillator signal. The oscillator circuit is shown in Figure 5a. The oscillator is a sawtooth waveform generator that employs current sources and a Schmitt trigger. The MOSFETs $M_2$ and $M_3$ are used as switches. A Schmitt trigger circuit with a NAND gate is employed to obtain a ramp signal and a clock signal. The signal clock and ramp signal are generated from a trigger signal.

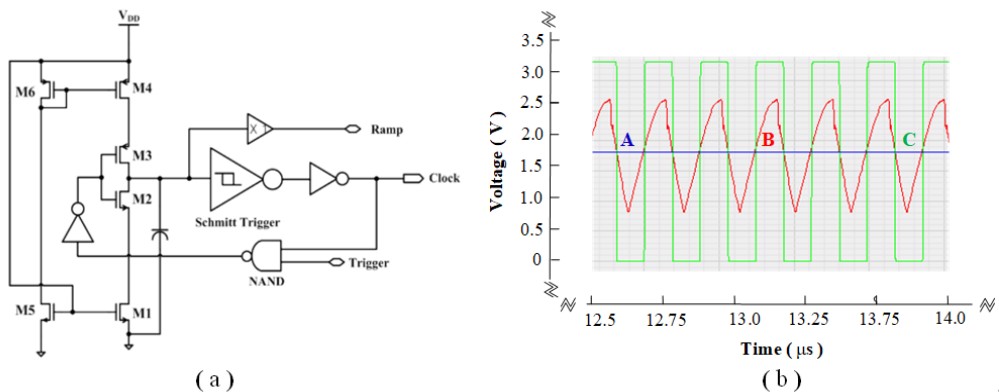

**Figure 5.** Proposed oscillator circuit: (**a**) oscillator circuit; (**b**) output of compensator (A), oscillator (B), and comparator (C).

Figure 5b shows the output waveforms of the compensator (A), oscillator (B), and comparator (C). When the compensator output voltage is less than the triangle waveform voltage, the output of the comparator is a low voltage. When the compensator output voltage is greater than the triangle waveform voltage, the output of the comparator is a high voltage.

## 2.4. Gate Driver Circuit

Figure 6a shows a gate driver circuit. The circuit is connected to power switches $MN_1$ and $MP_1$ mode of power MOSFETs. The driver circuit requires a careful design as it can draw large currents and generate overshoot currents during switching transitions. The

gate driver circuit consists of a NAND gate, two NOR gates, and two SR latches. The signals of the ramp (A) and clock (B) are shown in Figure 6b. The duty ratio, frequency, and amplitude can be changed by varying parameters of the oscillator.

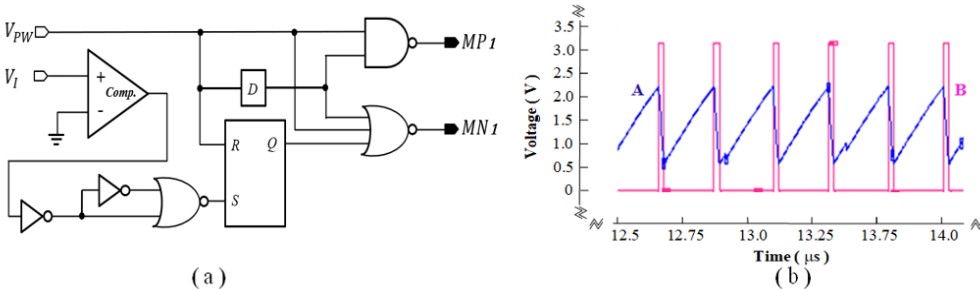

**Figure 6.** Proposed gate driver circuit: (**a**) gate driver circuit; (**b**) signals of ramp (A) and clock (B).

## 3. Results

### 3.1. Results of Simulation and Experiments

The voltage-mode DC–DC converter with an on-chip integrated PWM circuit operates at 5.5 MHz with an input voltage of 3.3 V. The power MOSFET in the power stage is isolated from the control stage to avoid noise. An *LC* filter is designed with an inductance of 1–10 µH and capacitance of 0.1–1 µF. The simulation result of inductor current $I_L$ and inductor voltage $V_L$ is shown in Figure 7a. It has been obtained with an input voltage of 3.3 V, inductance of 10 µH, capacitance of 1 µF, and duty ratio of 0.7. The charge and discharge times of the inductor current $I_L$ are accurately sensed. The ripple current depends on the output voltage, inductance, and duty ratio. The discharge time is approximately 0.07 µs.

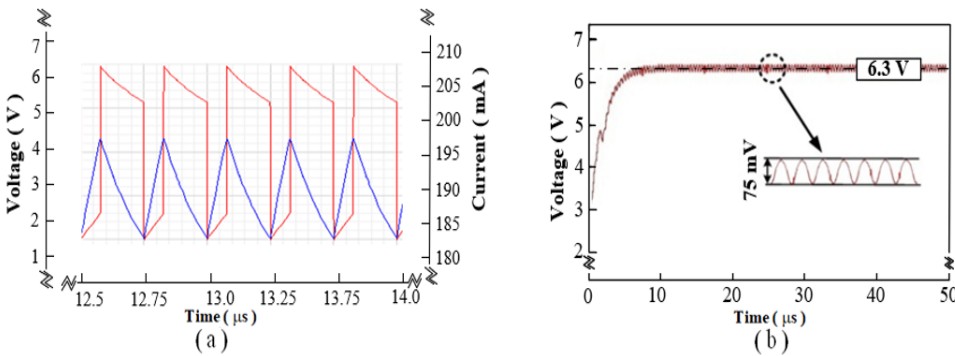

**Figure 7.** Simulation results: (**a**) output of inductor current $I_L$ (blue) and voltage $V_L$ (red) after inductor; (**b**) output voltage.

The inductor current is 190 mA within 5% ripple. Figure 7b shows the output and ripple voltages of DC–DC boost converter when they are 6.3 V and 75 mV, respectively. The output voltage has been obtained as expected with a negligible ripple ratio. In *LC* filter converters, glitches typically occur in the on-off switching transistors in the power stage. The Cadence simulation tool has been used to simulate the converter SoC of Figure 1. The duty cycle of the power switches $MN_1$ and $MP_1$ is approximately 70%.

Figure 8a shows the measured waveform of the input trigger signal and Figure 8b shows the measured waveform of the inductor voltage $V_L$, feedback voltage, and output voltage. The measured output voltage is 5.6 V at a switching frequency of 5.5 MHz. The voltage $V_L$ across the inductor shows a high ripple during on-off switching states. The output power is approximately 300 mW. The glitches in the (A), (B), and (C) waveforms are caused by a mismatch in the power switch of the $MN_1$ and $MP_1$ power MOSFETs. It was obtained with an inductor of 10 µH, input voltage of 3.3 V, capacitor of 1 µF, and duty ratio of 0.5.

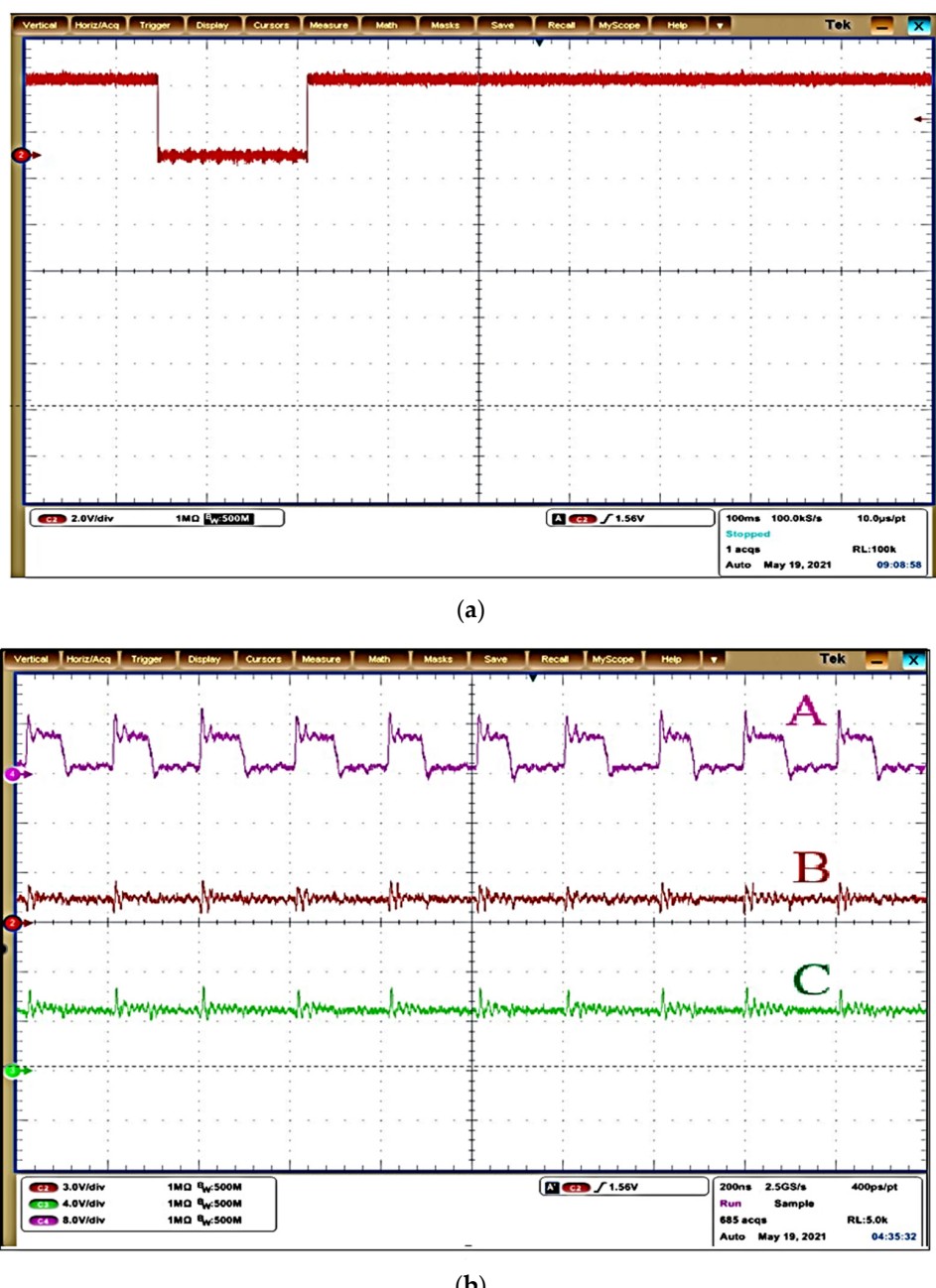

**Figure 8.** Experiments results: (**a**) input trigger signal; (**b**) inductor voltage $V_L$ after inductor (A), feedback voltage (B), and output voltage (C). (Scale 0.1 µs/div and 2.0 V/div).

Figure 9a shows the output voltage and efficiency with a variation of the load current. As the load current increases, the power efficiency is almost constant at 75%, and the output voltage decreases. This indicates that the efficiency is almost independent of the load current in the measured current range. However, when the load resistor becomes smaller with high frequency, the dynamic power loss is relatively larger than the static power loss, and the efficiency can be reduced because of the switching noise. The ripple voltage with a variation of the load current is shown in Figure 9b. The result corresponds to the dependency that can be obtained from Equation (5), and the ripple voltage is proportional to the load current. The ripple voltage is under 80 mV at the load current of 200 mA. Figure 9c shows the output voltage versus duty of the PWM signal. The output voltage linearly depends on the duty cycle, as expected.

$$V_{ripple} = \frac{I_{out}}{fC_{out}} \tag{5}$$

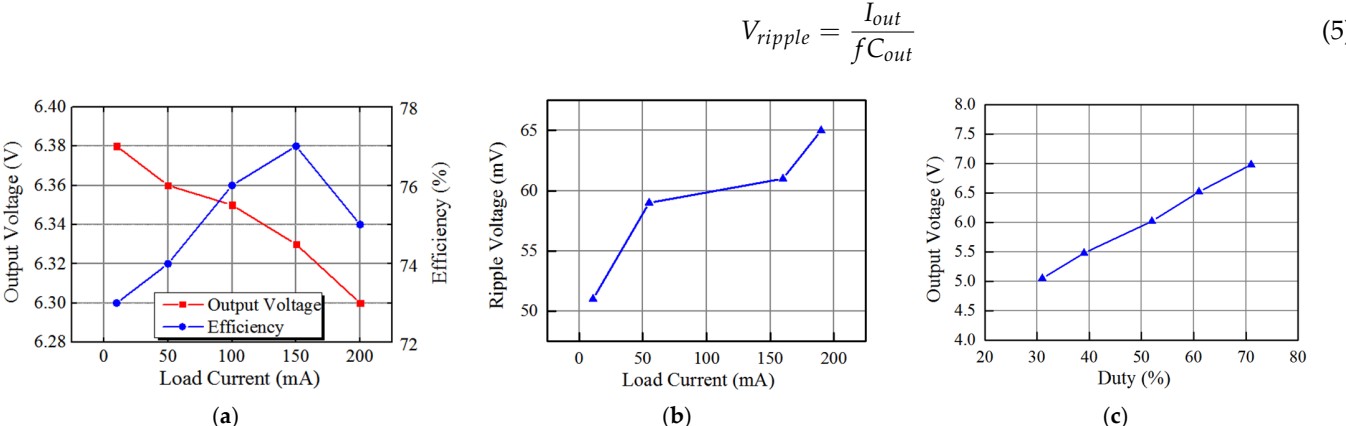

(a)

(b)

(c)

**Figure 9.** Test results: (**a**) output voltage and power efficiency with variation of the load current; (**b**) ripple voltage with variation of the load current; (**c**) output voltage versus duty.

### 3.2. Designed Condition and Photo Die of the Proposed DC–DC Boost Converter

The performance of the proposed voltage-mode DC–DC converter is summarized in Table 1. The proposed DC–DC converter achieves a high figure of merit with a small chip area.

**Table 1.** Performance summary.

| Type | Boost Converter |
|---|---|
| Process | 0.35 μm CMOS process |
| Input voltage ($V_{IN}$) | 3.3 V |
| Output voltage ($V_{OUT}$) | 5–7 V |
| Load current ($I_L$) | 15–200 mA |
| Switching frequency (*fs*) | 5.5 MHz |
| Ripple voltage | 10–100 mV |
| Efficiency | $\cong$75% (*I* mA) |
| Die area size | 0.5 mm$^2$ |

The proposed DC–DC boost converter is designed in 0.35 μm CMOS technology with 2-poly and 4-metal processes as shown in Figure 10.

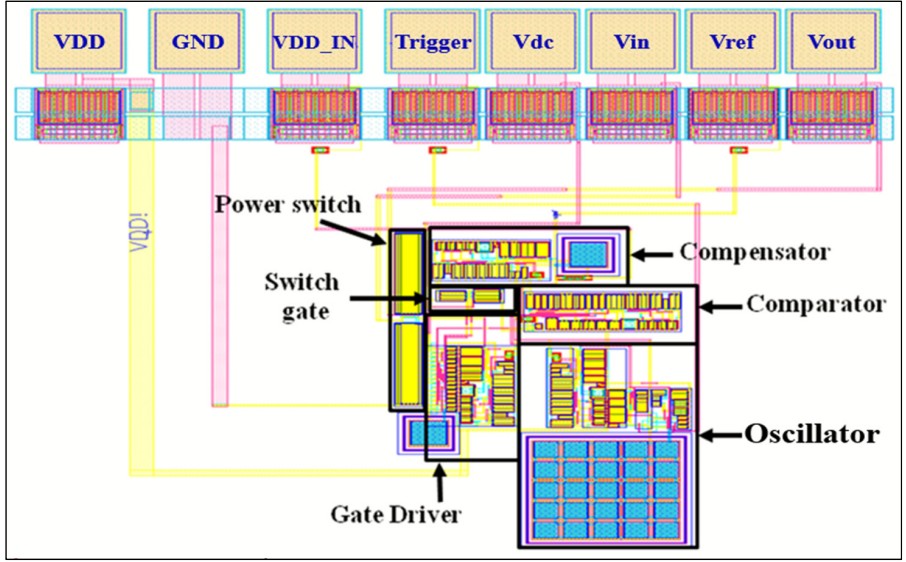

**Figure 10.** Die photo of the DC–DC boost converter.

## 4. Conclusions

A voltage-mode DC–DC boost converter with full integration is presented in this article. The proposed voltage-mode DC–DC boost converter is designed using 0.35-μm CMOS technology with 2-poly and 4-metal processes. Measurements of the fabricated converter have shown that the converter's output voltage ranges from 5–7 V with an input voltage of 3.3 V and a PWM switching frequency of 5.5 MHz. The feedback control circuit is applied in order to obtain a small ripple voltage that is almost independent of the load resistance. The feedback circuit operates with an OTA, amplifier, comparator, gate driver circuit, and oscillator. The proposed converter can be used as a miniaturized low-power LED driver IC. The power consumption has been found to be approximately 300 mW with maximum efficiency of $\cong$77%, which is applicable for high-performance LED display driver circuits. The proposed voltage-mode DC–DC boost converter offers such advantages as a simpler circuit structure than the current-mode converter and superior performance in terms of output voltage and power consumption.

**Author Contributions:** Conceptualization, C.-S.L., S.-G.C. and B.-C.A.; methodology, M.G. and C.-S.L.; validation, M.G., B.D. and C.-S.L.; formal analysis, C.-S.L., S.-G.C. and B.-C.A.; formal data curation B.D., S.-G.C. and C.-S.L.; writing—original draft, C.-S.L.; writing—review and editing, M.G., S.-G.C., C.-S.L. and B.-C.A. All authors have read and agreed to the published version of the manuscript.

**Funding:** This research was supported by Basic Science Research Program through the National Research Foundation of Korea (NRF) funded by the Ministry of Education, Science and Technology (grant number: NRF-2020R1I1A1A01066637), by the MSIT (Ministry of Science and ICT), Korea, under the Grand Information Technology Research Center support program (grant number: IITP-2022-2020-0-01462) supervised by the IITP (Institute for Information & communications Technology Planning & Evaluation), and by Basic Science Research Program through the National Research Foundation of Korea (NRF) funded by the Ministry of Education (grant number: 2020R1A6A1A12047945).

**Data Availability Statement:** Not applicable.

**Acknowledgments:** The authors would like to thank IDEC for the CAD tool support.

**Conflicts of Interest:** The authors declare no conflict of interest.

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
