# Peer review of "Integrated 0.35-µm CMOS Control Circuits for High-Performance Voltage Mode DC–DC Boost Converter"

_electronics, doi:10.3390/electronics12010133_

Round 1

Reviewer 1 Report (Previous Reviewer 1)

The paper presents an integrated boost converter made by 0,35 um CMOS technology. The revised version of the manuscript has higher quality and many mistakes were corrected. 

I have the following comments:

- please add references in line (35) after the sentence "... have extensively been studied."

- use small s instead of capital S for seconds in Fig. 7

- inductor voltage (A) in Fig. 8 can not have a DC value in a steady state. Now it has only a positive value and a nonzero average value.

I am also missing a more detailed analysis of the boost converter design ( components selection, MOSFET and drivers design, compensator design, etc.). But it is probably out of the scope of this paper. The paper seems more like a letter, not an article.

Author Response

Dear Reviewer

I appreciate the reviewer's comments on my manuscript.

My co-authors and I have revised the manuscript to reflect the reviewer's comments.

The simulation data and experimental data are attached to the manuscript as shown below.

 Thank you again for your kind consideration.

regards,

Lee Chan-soo

Reviewer 2 Report (New Reviewer)

I congratulate the authors for the excellent work, but several corrections should be made to improve the paper.

1.. Abstract must be improved to consider the problem, objective, methodology, findings, and main contribution and results.

2.. It is necessary to improve the state of the art with papers related to the research topic. In the introduction, it is necessary to make a good state of the art with recent papers, with research related to the objectives of this paper.

3.. The statement of the problem and the objectives of the research are not clear.

4.. A list of contributions of the paper should be made, please add it at the end of the introduction.

5.. In the experimental results, show the time axes to check if you are switching at 5 MHz

6.. A space is needed on line 75 before the parentheses, make a general review of this type of error in the document.

7.. All figures should be improved in quality and sharpness.

8.. In the name of the figures at the bottom, the journal format is not being followed, the letter (b) should not be placed at the end.

9.. In the results, a test of load changes must be made at the simulation level and experimentally.

10.. Figure 9 has poor quality

11.. The results should show the behavior for different changes in duty cycle.

12.. Please show a complete diagram of the simulation.

13.. It is necessary to show the mathematical models or some formula that supports this investigation.

14.. Improve the Table 1 according to the format of the journal.

15.. Finally, more tests and results are needed to better support the investigation.

16.. The conclusions should be improved, with respect to the objectives and contributions of the paper.

17.. Matches with other papers are attached to avoid repetition.

Author Response

Dear Reviewer

I appreciate the reviewer's comments on my manuscript.

My co-authors and I have revised the manuscript to reflect the reviewer's comments.

My manuscript was limited to 9 pages for this journal. please understand. 

The simulation data and experimental data are attached to the manuscript as shown below.

Thank you again for your kind consideration.

regards,

Lee Chan-soo

Round 2

Reviewer 2 Report (New Reviewer)

The authors did not make the requested corrections, I am attaching again the tasks that must be done to accept the paper.

I confirm that the Electronics (ISSN 2079-9292) journal  has no limit on the number of pages.

1.. Abstract must be improved to consider the problem, objective, methodology, findings, and main contribution and results.

2.. It is necessary to improve the state of the art with papers related to the research topic. In the introduction, it is necessary to make a good state of the art with recent papers, with research related to the objectives of this paper.

3.. The statement of the problem and the objectives of the research are not clear.

4.. A list of contributions of the paper should be made, please add it at the end of the introduction.

5.. In the experimental results, show the time axes to check if you are switching at 5 MHz

6.. A space is needed on line 75 before the parentheses, make a general review of this type of error in the document.

7.. All figures should be improved in quality and sharpness.

8.. In the name of the figures at the bottom, the journal format is not being followed, the letter (b) should not be placed at the end.

9.. In the results, a test of load changes must be made at the simulation level and experimentally.

10.. Figure 9 has poor quality

11.. The results should show the behavior for different changes in duty cycle.

12.. Please show a complete diagram of the simulation.

13.. It is necessary to show the mathematical models or some formula that supports this investigation.

14.. Improve the Table 1 according to the format of the journal.

15.. Finally, more tests and results are needed to better support the investigation.

16.. The conclusions should be improved, with respect to the objectives and contributions of the paper.

Author Response

Please find the attached authors' reply to the reviewer comments. We have made some non-small efforts to improve the manuscript according to the reviewer's comments. Thanks.

Round 3

Reviewer 2 Report (New Reviewer)

Thank you very much for the corrections made.

To improve authors should do the following:

The introduction must be improved, it is necessary to improve a little more in the state of the art with more paper related to the research topic, the problem statement and the objectives must be made clear, at the end make a list of the contributions of the paper.

Finally if it is possible to place a photo of the experiment labeling the parts of it.

Best regards

Author Response

We appreciate reviewer's comments.

We have tried to improve the manuscript as much as possible reflecting reviewer's comments.

We understand that out manuscript has been significantly improved with reviewer's helpful comments.

Thanks.

This manuscript is a resubmission of an earlier submission. The following is a list of the peer review reports and author responses from that submission.

Round 1

Reviewer 1 Report

The paper presents the design of the integrated boost DC/DC converter intended to supply LEDs. 

The introduction is a bit short and could be extended by analysing the current state of the art and problems in the area of integrated DC/DC converters with sub 1 W output power. This would highlit the novelty of the proposed design. However, I see that the paper is aimed more at the description of the design procedure. 

Put a space before the bracket (e.g. lines 146, 147, 148, 150, 153, etc.)

Describe Fig. 7 in more detail. Fig. 7 (a) is not a circuit but some waveforms and there are no (A) and (B) signals in Fig. 7 (b). Also, you have mentioned the inductor current value to be 190 mA in the text, however, there is 1,9 A in Fig. 7 (a). 

The practical results in Fig. 8 are unreadable. How is the input trigger signal related to the output signal? It is impossible to read the values from oscillograms. 

Fig. 9 has low quality and part of it is upside down.

There are several mistakes in the text, e.g. on lines 64, 99, 115. The text needs to be edited.  

Reviewer 2 Report

No Comments

Reviewer 3 Report

The following critical questions have not been presented in the current version. It is needed to revise the manuscript significantly.

1. What are the issues to be addressed by the authors?

2. What circuit blocks have been updated over the existing designs?

3. How much does the proposed design improve what aspects over the existing designs?

The manuscript must be proofread. A few examples are

- In abstract, "Low voltage operation is on of intrinsic circuit characteristics" must be "Low voltage operation is one of intrinsic circuit characteristics".

- In Introduction, "The DC-DC converter reduce a power transistor" must be "The DC-DC converter reduces a power transistor"

- In Introduction, what do the authors mean by "to the power consumption"?

- In Introduction, "By using the properties of CMOS transistor" is mentioned. What properties do the authors mean?

- In Introduction, the authors describe "there are a lot of difficulties for circuit realization", but there is no explanation on what difficulties.